# Australian Aboriginal and Torres Strait Islander Health Information: Progress, Pitfalls, and Prospects

**DOI:** 10.3390/ijerph181910274

**Published:** 2021-09-29

**Authors:** Ian Ring, Kalinda Griffiths

**Affiliations:** 1Tropical Health and Medicine, James Cook University, Townsville 4810, Australia; 2Centre for Big Data Research in Health, Medicine, University of New South Wales, Sydney 2052, Australia; kalinda.griffiths@unsw.edu.au

**Keywords:** Aboriginal and Torres Strait Islander health, Indigenous health measurement, life expectancy, misleading statistics, management use of information, data sovereignty, governance

## Abstract

Despite significant developments in Aboriginal and Torres Strait Islander Health information over the last 25 years, many challenges remain. There are still uncertainties about the accuracy of estimates of the summary measure of life expectancy, and methods to estimate changes in life expectancy over time are unreliable because of changing patterns of identification. Far too little use is made of the wealth of information that is available, and formal systems for systematically using that information are often vestigial to non-existent. Available information has focussed largely on traditional biomedical topics and too little on access to, expenditure on, and availability of services required to improve health outcomes, and on the underpinning issues of social and emotional wellbeing. It is of concern that statistical artefacts may have been misrepresented as indicating real progress in key health indices. Challenges and opportunities for the future include improving the accuracy of estimation of life expectancy, provision of community level data, information on the availability and effectiveness of health services, measurement of the underpinning issues of racism, culture and social and emotional wellbeing (SEWB), enhancing the interoperability of data systems, and capacity building and mechanisms for Indigenous data governance. There is little point in having information unless it is used, and formal mechanisms for making full use of information in a proper policy/planning cycle are urgently required.

## 1. Progress

Thompson [1] and Smith [2] have described the early history of the development of Aboriginal and Torres Strait Islander health statistics. In brief, the National Health and Medical Research Council (NHMRC) in 1955 drew attention to the fact that despite reported high levels of Indigenous morbidity and mortality in parts of Australia, precise information was not available. The first regular collection of data was commenced by the Northern Territory (NT) administration on infant mortality in 1957, but that was the only systematic collection for many years. In 1973 Commonwealth and jurisdictional Health Ministers endorsed a policy of collecting national Aboriginal health statistics. Progress was painfully slow and in the early 1980s no jurisdiction identified Aboriginal and Torres Strait Islander people in birth and death records. This is despite the 1967 constitutional changes to include Aboriginal and Torres Strait Islander people in the national population count. In 1984 the Commonwealth established a high-level taskforce on Aboriginal health statistics, but progress with implementation of its recommendations to prioritise Indigenous identifiers in vital statistics and hospital and perinatal statistics by the jurisdictions was patchy. Responsibility then passed to the newly formed Australian Institute of Health (AIH), but the funds provided for progressing the development of Indigenous health statistics were only half of those recommended arising from the National Aboriginal Health Strategy (NAHS) in 1989.

There have been extensive developments in the capture of Aboriginal and Torres Strait Islander data for the purposes of national statistics in the last 60 years. Since federation, there have been a number of laws enacted for the purposes of identifying and counting Aboriginal and Torres Strait Islander people [3]. While there are departments, centres, and groups within the Australian government that focus on Aboriginal and Torres Strait Islander statistics, there have been some, but limited, developments in government support for Aboriginal and Torres Strait Islander oversight. Historically, it has often been individuals within government who have worked with Aboriginal and Torres Strait Islander communities and individuals to support the visibility of Indigenous people in the nation. 

The Australian Bureau of Statistics (ABS) and the Australian Institute of Health and Welfare (AIHW) instituted a joint unit in Darwin in 1996. In 1997 this unit produced the first in a series of what was intended to be flagship biennial publications on the Health and Welfare of Australia’s Aboriginal and Torres Strait Islander Peoples [4]. Importantly, the first edition was launched in Darwin by the Governor General of Australia, Sir William Deane, who emphasised the importance of good statistics to drive good policy and action:

“This report will hopefully do much to influence all Australians, both Indigenous and non-Indigenous, to approach the question of the health and welfare of Aboriginal and Torres Strait Islander peoples, particularly children, on the basis of unprejudiced statistical facts [5].”

The joint unit was disbanded after 7 years, and ABS and AIHW followed independent paths. 

### 1.1. The Aboriginal and Torres Strait Islander Health Information Plan

It was in this context that the Aboriginal and Torres Strait Islander Health Information Plan was prepared for the Australian Health Ministers Advisory Council (AHMAC) by ABS and AIHW in 1997 [6], appropriately subtitled *…This time let’s make it happen*, and is a convenient starting point. The subtitle is an explicit recognition of the relative failure of previous attempts to make significant progress with this important topic. As the foreword to the report says: 

In 1994 the AHMAC endorsed the recommendation of the national body responsible for national health information, that the highest national priority was to:

“Work with Aboriginal and Torres Strait Islander peoples to develop a plan to improve all aspects of information about their health and health services.”

Funds were provided to implement that recommendation and develop a plan.

AHMAC accepted the recommendations of the Report and instructed the National Health Information Management Group (NHIMG) to oversee the implementation. NHIMG established an implementation group including Indigenous health organisations and other agencies for this purpose.

The report described the shortcomings in the collection, processing and use of Indigenous health information, and emphasized the central role of the poor quality of Indigenous identification in current collections. The report went on to say that there was little new in its findings and recommendations and noted the lack of commitment to implement the findings of the numerous reviews that had been undertaken as the chief reason for the overall lack of progress.

Up until the publication of this report, the main source of national information had been the National Aboriginal and Torres Strait Island Survey conducted by the ABS in 1994 [7]. This was as part of the Government’s response to Recommendation 49 [8] of the Royal Commission into Aboriginal Deaths in Custody, “That proposals for a special national survey covering a range of social, demographic, health and economic circumstances of the Aboriginal population with full Aboriginal participation at all levels be supported”. The aim was to provide Australian governments with a “stronger information base for planning for the empowerment of Australia’s Indigenous peoples and for measuring progress in meeting their objectives, aspirations and needs”.

### 1.2. National Advisory Group on Aboriginal and Torres Strait Islander Health Information and Data (NAGATSIHID)

NAGATSIHID was established “as a result of a decision by AHMAC in October 2000, to improve reporting on the health status of Indigenous Australians. It was set up as the national body to create a partnership between the Commonwealth, jurisdictions and Aboriginal and Torres Strait Islander people to improve Indigenous information in national and jurisdictional data collections” [9]. The purpose of the committee was to make strategic decisions regarding the use of government held data pertaining to Aboriginal and Torres Strait Islander people and work to improve the quality and accessibility of Indigenous data and information.

What made NAGATSIHID different from other committees was: “(i) the level of representation from the governments (chaired by an AHMAC member); (ii) it had a majority Aboriginal and Torres Strait Islander membership with representatives from a wide range of key stakeholders in Aboriginal and Torres Strait Islander health such as the community controlled sector, academia and the government sector with decision making made through an Aboriginal and Torres Strait Islander quorum; (iii) it provided a unique example of an effective working partnership between government agencies, Aboriginal and Torres Strait Islander people and organisations to advance the development and use of data and information on the health of Indigenous Australians; (iv) having a majority of Indigenous people on NAGATSIHID gave the agencies some confidence that the decisions by AHMAC (through NAGATSIHID) reflect the views of Indigenous people and their representative bodies; and (v) it is recognised internationally and has been responsible for many of the significant changes in Aboriginal and Torres Strait Islander health statistics and data” [9].

The main role of NAGATSIHID was “to provide broad strategic advice to AHMAC, and in particular was responsible for: •Continuing the implementation of the 1997 Aboriginal and Torres Strait Islander health Information Plan—*this time let’s make it happen* (AIHW 1997 [6]);•Advising AIHW and ABS on information and data priorities;•Providing advice to the Australian Government’s Department of Health (DoH) on the Aboriginal and Torres Strait Islander Health Performance Framework (HPF)” [10].

NAGATSIHID was abolished in 2019, without notice to its members. While there are a number of advisory committees within government agencies [11,12] to support decision-making regarding Aboriginal and Torres Strait Islander data by those individual agencies, there has been no replacement to the principal committee.

### 1.3. Development of National Surveys

In 2001, the National Health Survey was enhanced with a supplementary sample of Indigenous people of sufficient size to produce national estimates for Indigenous people. The supplementary sample was part funded by the Commonwealth and the jurisdictions, and became the first national Indigenous health survey. This was followed by a larger supplementary Indigenous sample in 2004 to provide both national and jurisdictional estimates, and thereafter, was conducted every 6 years [13].

Even though a national biomedical risk factor survey had been conducted for the Australian administration in Papua New Guinea in the late 1960s [14], it was not until 2012–2013 that a parallel survey was conducted in Australia, and was made possible by additional funding provided by the Australian Government Department of Health and the National Heart Foundation of Australia. This national survey included two Indigenous components, a National Aboriginal and Torres Strait Islander Nutrition and Physical Activity Survey and a National Aboriginal and Torres Strait Islander Health Measures Survey [15].

### 1.4. ABS and AIHW Publications

Currently, the ABS has a range of publications concerning Aboriginal and Torres Strait Islander peoples, based largely on the census and the extensive ABS survey program [13] covering health surveys; population estimates and projections; life tables; understanding the increase in census counts; Torres Strait Islander people characteristics; Aboriginal and Torres Strait Islander women; smoking trends; education, etc.

The AIHW regularly produces a wide variety of publications on Indigenous health and welfare topics. Recent topics include: the Health Performance Framework; acute rheumatic fever and rheumatic heart disease; Indigenous injury deaths; Indigenous specific primary health care datasets: The Online Services Report and the national Key Performance Indicators; Northern Territory remote Aboriginal investment: oral health program; better cardiac care measures for Aboriginal and Torres Strait Islander people; cultural safety in health care for Indigenous Australians; hearing health outreach services for Aboriginal and Torres Strait Islander children in the Northern Territory; aged care; disability support; Indigenous community safety; Indigenous education and skills; Indigenous employment; Indigenous housing; Indigenous income and finance; understanding Indigenous welfare and wellbeing; Indigenous eye health; Indigenous mental health and suicide prevention clearinghouse, etc.

### 1.5. The Overcoming Indigenous Disadvantage (OID) Report

The Council of Australian Governments (COAG) commissioned the OID [16] report in 2002, and nominated two core objectives for the report:•To inform Australian governments about whether policy programs and interventions are achieving improved outcomes for Aboriginal and Torres Strait Islander people;•To be meaningful to Aboriginal and Torres Strait Islander people.

As the 2020 report [17] says, “This edition of the report seeks to identify the significant strengths of, and sources of wellbeing for, Aboriginal and Torres Strait Islander people—and to illustrate the nature of the disadvantage they experience, focusing on the key structural and systemic barriers that contribute to this disadvantage. The framework of indicators focuses on some of the factors that contributed to their wellbeing or that cause the disadvantage they experience, these factors were selected based on evidence, logic and where experience suggests that targeted policies will have the greatest impact. The indicators are supplemented by additional research on structural and systemic barriers that contribute to, or maintain, the disadvantage experienced by Aboriginal and Torres Strait Islander people, and where governments may have a role in removing barriers.”

### 1.6. The Aboriginal and Torres Strait Islander Health Performance Framework

The purpose of this report is said to be that “The Aboriginal and Torres Strait Islander Health Performance Framework (HPF) monitors progress in Aboriginal and Torres Strait Islander health outcomes, health system performance and the broader determinants of health (such as employment, education and safety). The HPF is a comprehensive source of evidence designed to inform policy, planning, program development and research.

Beginning in 2006, HPF reports have been released every 2–3 years. The HPF includes data analysis drawn from over 60 data collections, findings from research and evaluations, and analysis of implications of the evidence for government, health services and the research sector.

The HPF consists of 68 measures across three domains (Tiers): Tier 1—Health status and outcomes; Tier 2—Determinants of health; Tier 3—Health system performance” [18].

### 1.7. Expenditure

The Indigenous Expenditure Report (IER) aims to contribute to better policy making and improved outcomes for Indigenous Australians and will

“3. include expenditure by both Commonwealth and State/Territory governments (and local government if possible), and over time will: (a)Allow reporting on Indigenous and non-Indigenous social status and economic status;(b)Include expenditure on Indigenous-specific and key mainstream programs;(c)Be reconcilable with published government financial statistics.

4. focus on on-the-ground services in areas such as: education; justice; health; housing; community services; employment; and other significant expenditure

6. provide governments with a better understanding of the level and patterns of expenditure on services which support Indigenous Australians, and provide policy makers with an additional tool to target policies to Close the Gap in Indigenous Disadvantage” [19].

Reports have been produced periodically since 2010 with the most recent report being the 2017 version.

While the IER produced by the Productivity Commission focuses on government expenditure, expenditure analysis carried out by AIHW for the HPF “encompasses government, non-government, private and individual expenditure on health and medical services, hospital services (admitted and non-admitted patients), community health services, dental services, aids and appliances, pharmaceuticals, patient transport and public health programs…”. It points out that “four interacting factors within Australia’s health system potentially have major consequences for the health of many Aboriginal and Torres Strait Islander people, namely limited Indigenous-specific primary health care services; Indigenous Australians’ underutilisation of many mainstream health services and limited access to government health subsidies; increasing price signals in the public health system (such as co-payments) and a low Indigenous private health insurance rate; and failure to maintain real health expenditure levels over time” [20]. An important element of the AIHW expenditure analysis is that, unlike the IER, it includes non-government expenditure as well as government expenditure, allowing for a more meaningful comparison of Indigenous and non-Indigenous expenditure on health and social areas.

### 1.8. Indigenous Data Developments

A range of conversations and meetings to identify what is required for data to support the needs and aspirations of Aboriginal and Torres Strait Islander people have occurred more frequently over the past 5 years. Emerging from these discussions, Aboriginal and Torres Strait Islander people identified the need for strategic government and organisational partnerships to work towards the development of the data capabilities of Aboriginal and Torres Strait Islander communities for the purpose of community advancement.

Initiatives such as the Maiam nayri Wingara Indigenous Data Sovereignty Collective [21] and the Indigenous Data Network (IDN) [22] have emerged as Indigenous-led groups to support the systems and governance of Indigenous data. Further, there has been a range of advocacy and negotiations between Aboriginal and Torres Strait Islander leaders and governments to further develop Indigenous data, particularly at the regional level. Recently, the IDN was funded by the Australian Government via the National Aboriginal Community Health Organisation (NACCHO), and is a part of the National Agreement on Closing the Gap [23] (National Agreement), which focuses on shared access to data and information at a regional level.

The $1.3 million project, led by Indigenous researchers and experts from around the country, was to support Priority Reform Four of the National Agreement that aims to improve and share access to data and information to enable Aboriginal and Torres Strait Islander communities to make informed decisions.

The IDN had been working in partnership with the Coalition of Aboriginal and Torres Strait Islander community-controlled peaks (Coalition of Peaks) to support the development of a new platform, which will enable Indigenous organizations to upload and analyze their own data.

“The data collected will be focused on the areas and targets, including the Priority Reforms, in the newly agreed National Agreement on Closing the Gap. It will span health, education, employment, justice, environmental management and cultural heritage services, ensuring Indigenous organisations can make evidence-based decisions to set strategies that are aligned to community needs”.

“The launch of this project is the latest achievement for the IDN, which was established in 2017 to give voice to the principles of Indigenous data sovereignty—the recognition of intellectual property and other rights [24] of Indigenous people and entities in their data so that it cannot be harvested without consent by governments or any other data collector—and to lead a push for the implementation of national Indigenous data governance framework” [22].

Announcing the data project in his 2021 February address, Closing the Gap Statement to Parliament, Prime Minister Scott Morrison said that “a vital part of empowering Indigenous communities is giving them the data and information to inform their decision making.” [25].

### 1.9. Data Sharing

The Australian government has invested significantly in its national data capabilities to monitor the progress of the nation’s health through data sharing. In August 2018, the Prime Minister and Cabinet established the Office of the National Data Commissioner to build and support the infrastructure and use of public data [26]. Other national initiatives have included the National Collaborative Research Infrastructure Strategy and the Strategic Committee for National Health Information to make better use of research and health data [27]. These initiatives are developments arising from a range of internal government developments in data sharing. This includes the ABS Multi-Agency Data Integration Project (MADIP) in 2015. After its establishment, almost $131 million was invested in the Data Integration Partnership for Australia (DIPA) from 2017 to 2020 to improve technical data infrastructure and data integration capabilities across the Australian Public Service. These data assets have and continue to be used for a range of government projects and have the potential to improve statistical understandings as well as data quality.

For Aboriginal and Torres Strait Islander people there are limited mechanisms to govern Indigenous data within governments. There is currently no available information regarding who is making decisions regarding linked Aboriginal and Torres Strait Islander data and the above-mentioned data assets. In terms of data sharing, there is still a way to go regarding the interoperability of the data systems and platforms outside of government. This includes the linkage of primary health care, disease registries, and surveillance systems, and broader sectors of data collections, such as education and justice, which can provide critical insights to the distribution and determinants of health and disease in Australia.

### 1.10. International Indigenous Information Developments

The International Group for Indigenous Health Measurement [28] (IGIHM) was founded in 2005 and brings together Indigenous and non-Indigenous, government and non-government, statisticians, researchers, and health professionals from the four founding members of this group, Australia, Canada, New Zealand, and the United States, and, more recently, representatives from Sami organizations and Indigenous peoples from South America. The IGIHM’s goals are “first, to promote awareness of the deficiencies of health data for Indigenous populations in our four countries and second, to collaborate internationally on improved methods and policies that will contribute to the improvement of Indigenous health. Since its founding in 2005, the IGIHM has pursued a variety of activities to further its goals. These activities have centred on multi-national partnerships as well as the promotion of improved methods for the collection, analysis, interpretation and dissemination of information useful for improving the health of Indigenous populations, enhancing Indigenous health knowledge and data, and the elimination of health disparities” [29]. A major recent focus has been on promoting Indigenous measurement issues in international forums including UN Statistical agencies, and the International Association for Official Statistics.

## 2. Pitfalls

### 2.1. Census

In the 1967 Referendum, Australians voted overwhelmingly to amend the Constitution to allow the Commonwealth to make laws for Aboriginal people and include them in the census. “Turnout for the referendum was almost 94 per cent, and the result was a strong ‘Yes’ vote, with a significant majority in all six states and an overall majority of almost 91 per cent…” [30]. The legislation for the referendum was passed unanimously by the parliament.

The ABS had compiled experimental life tables for Indigenous Australians following the 1996 and 2001 Censuses of Population and Housing. Those estimates were compiled using different indirect demographic methods and were subject to a range of caveats [31]. Subsequently, ABS changed its methodology to direct methods. This change in method was generally welcomed although it was argued that the direct method understated Indigenous deaths and overstated life expectancy [32].

The direct method attempts to correct for under identification of deaths by use of the Post-Enumeration Survey (PES), but there is some uncertainty about the accuracy of national estimates for Indigenous life expectancy as the PES may be too small in the 60+ group, leading to high raising fractions based on small numbers of deaths, and there is also uncertainty about the adequacy of the size of the linked deaths/census sample itself. Further, the fact that ABS and AIHW produce similar estimates for life expectancy using different methods, rather than adding weight to the accuracy of both, suggests that both may overstate life expectancy as the AIHW method [33] is based on data sources, all of which are known to be incomplete.

Apart from the concerns about the accuracy of national estimates of Indigenous life expectancy derived from the census, the capacity to detect differences between successive five-yearly national life expectancy estimates, as statistically significant is at best doubtful [34]. This is in part because of significant changes in Indigenous identification between successive censuses. It is estimated that between 2011 and 2016 approximately 120,000 people who identified as non-Indigenous in 2011 identified as Indigenous in 2016, and approximately 40,000 people who identified as Indigenous in 2011 identified as non-Indigenous in 2016 [35]. Thus, a net 80,000 people changed identification from non-Indigenous to Indigenous from a census count of approximately 650,000 in 2016 and these newly identified people largely lived in cities and were better educated, more likely to be employed and had higher incomes—and were presumably healthier. Given the potential errors in each census and the proportionate size of the change in identification (approximately one in 8) and the fact that the newly identified people may well have been healthier, it becomes difficult if not impossible to determine whether any apparent increase in life expectancy between successive censuses is real or at least partially due to statistical artefact.

### 2.2. Backcasting

The other main method in assessing the extent of mortality or other changes over time is by the use of backcasting. “This technique requires assumptions to be made about past levels of mortality taking into account the most recent 2016 census data to utilise the best quality estimates available. These are applied to the 2016 base population to obtain a ‘reverse-survived’ population for the previous year. The assumptions are then applied to this new reverse-survived population to obtain a population for the preceding year. This process is repeated until the first year of the estimation period is reached [36].” ABS provides backcast population estimates for 2006–2015, but advises caution in backcasting for earlier periods:

“ABS advises that the 2001 to 2005 estimates included in the spreadsheet attached to this release should be used with caution. 

Reliable life expectancy estimates of the Aboriginal and Torres Strait Islander population are not available for the period 2001 to 2005. Therefore, mortality assumptions for these years were based on trends in life expectancy during 2005–2007 and 2015–2017. There will be a greater alignment between this assumption-based mortality and the actual mortality for the years closer to the base year than those for the out years.

Moreover, estimates of the Aboriginal and Torres Strait Islander population on 30 June 2016 (based on the 2016 census) are 19% larger than those on 30 June 2011 (based on the 2011 Census). As a consequence, the use of this 2016 ERP base introduces uncertainty to the historical estimates. The uncertainty increases as the time from the base year increases”.

Apart from the historical uncertainty about population estimates for earlier periods as noted by the ABS, there is a troubling circularity in the method in that in estimating trends in mortality, the method is dependent on assumptions about the mortality trends—the very parameter being estimated.

Nonetheless, government agencies show mortality trend graphs going as far back as 1998 [17,18,37]. A typical graph is shown below [18] (Figure 1). The commentary accompanying the graph says that “these changes resulted in the gap between the two populations decreasing significantly by 49% from 1998 to 2018. Most of this improvement was seen between 1998 and 2006, when the gap narrowed significantly by 42%. Over the period 2006 to 2018, the gap continued to narrow by 8% but this was not a significant change.”

It is hard, if not impossible, to explain what health service, social, economic, or political changes might plausibly account for such dramatic improvements (42%) in the mortality gap between 1998 and 2006, and at the same time for a non-significant change in the mortality gap between 2006 and 2018. It is quite possible, and perhaps likely, that the apparent dramatic improvements between 1998 and 2006 were statistical artefacts associated with a lack of attention to the ABS cautionary advice, rather than real changes.

It might reasonably be concluded that it is unsafe to backcast for longer than 10 years. On that basis, the AIHW conclusion is that, “Consistent with the observed decline in mortality, life expectancy at birth increased for both Indigenous males and females during the reference period (2001–2005 to 2011–2015). However, greater increases in life expectancy at birth occurred for non-Indigenous males and females, meaning that the gap in life expectancy between Indigenous and non-Indigenous Australians widened during the reference period” [33]. This conclusion may provide the most reliable view of trends in life expectancy in recent years.

### 2.3. Misleading Use of Statistics

In addition to the technical issues outlined above, the most recent example of misleading use of statistics can be found in the Productivity Commission’s 2021 Closing the Gap report [38].

The text here and elsewhere in the report says that this indicator (healthy birthweight) is “on track”. This is manifestly not the case and is apparently based on just two points, 2017 and 2018. Projecting a trend from two points is simply inappropriate as the accompanying graph and supporting tables makes clear and the caveat does not deal with the real issue—the indicator is actually not on track. Many readers may struggle to reconcile the graph, and the commentary below it (Figure 2), indicating that there has been no change in the indicator, nationally or for any of the jurisdictions, with the assertion that the indicator is on track. A reasonable commentary based on the available information might have read, “There is insufficient data since the baseline year (2017) on which to base a trend, but the period from 2014 to 2018 does not suggest the target is on track.”

The interests of First Nations peoples are in no way served by asserting that such a key indicator is on track and hence current efforts to improve the health of mothers and infants are adequate, when, in reality, that is far from being the case and significantly greater effort is required so that this key indicator will cease to flatline and will start to move in the right direction. The material from the report for this indicator is shown above.

### 2.4. Use of Information

Though there were and are limitations in the data that were available and some significant gaps in available information, for many years now, there has been a wealth of information that was available and readily accessible to administrators and policy makers. The Aboriginal and Torres Strait Islander Health Performance Framework, the Health and Welfare of Australia’s Aboriginal and Torres Strait Islander Peoples, the OID Reports, and the annual Closing the Gap Reports released by the Prime Minister at the opening of parliament each year all provide a wealth of information on health indices and progress or lack of progress.

Yet there seems no formal process for policy makers and service providers to examine each report and take policy and management decisions on the findings. The process seems little better than sitting around hoping next year’s numbers might look better without taking formal action to evaluate the findings and take the necessary action to improve performance—particularly in a climate where all can see that progress has been inadequate. This is in part because: evaluation is generally bitty, piecemeal, and not embedded in a formal policy and planning cycle; in part because the sheer volume of material makes it almost indigestible; a false sense of reassurance compounded by misrepresentation of statistical artefact as real progress; too little information is available on the availability of, funding for, access to, appropriateness or effectiveness of services required to improve outcomes; also because indicators are reported on as discrete measures separately and independently and the interrelationship between them not specified (if progress in all causes mortality is disappointing, no information is provided on services for chronic disease); information is generally only available at national and jurisdictional levels rather than service delivery or community levels; but above all there is simply no formal process to examine the content of these reports and see what lessons could and should be learnt to achieve the progress specified in national goals.

Monthly, six monthly, and annual reviews to examine available data on performance are not a feature at any level, certainly not at national, jurisdictional, or regional levels, though some services may be doing so. This is amateur hour writ large—and the consequences for the health and welfare of Aboriginal and Torres Strait Islander people are very significant in terms of preventable admissions and deaths.

### 2.5. Surveillance and Monitoring Services

There is a pressing need to ensure that disease surveillance systems and service monitoring continue to be efficient, effective, and appropriate to enable timely and appropriate services to the public. This includes communicable and non-communicable diseases as well as primary health care services. Perhaps the most immediate issue impacting Aboriginal and Torres Strait Islander people is the absence of the Indigenous identifier on private pathology request forms. This affects measurement of many issues, cancer, infectious diseases, and currently COVID-19. When knowledge of testing rates is so critical to the prevention and management of COVID-19 and with Indigenous people at particular risk, it is hard to believe that this most crucial piece of information is still lacking despite numerous calls for improvement. Most recently, the National Aboriginal and Torres Strait Islander COVID-19 Management Plan [39] had the recommendation for a remit to improve data collection and Aboriginal and Torres Strait Islander identification in healthcare and pathology testing.

More generally, notwithstanding the 1994 AHMAC decision that the highest national health information priority was to “work with Aboriginal and Torres Strait Islander peoples to develop a plan to improve all aspects of information about their health and health services [6]”, most of the subsequent development work on Indigenous health information has centred around health rather than health services, although AIHW has done some useful work in this area [40,41] and the HPF and the OID Reports provide some basic information. Nonetheless, there is little essential information available on service gaps (which could, for example, be defined as areas with high levels of preventable admissions and deaths and low use of the Medical and Pharmaceutical Benefits Schedules BMBS/PBS) and less about how well services that do exist actually work. The Productivity Commission found that, “There are many Australian Government policies and programs that are designed to improve the lives of Aboriginal and Torres Strait Islander people. But after decades of developing new policies and programs and modifying existing ones, we still know very little about their impact on Aboriginal and Torres Strait Islander people, or how outcomes could be improved [42].” This has been a serious omission as it has meant that much information has been provided about health issues for Aboriginal and Torres Strait Islander peoples, but not the kind of information which would provide policy makers and administrators with the information required for more effective action to address those health issues.

## 3. Prospects

### 3.1. Life Expectancy

Progress with some of the issues outlined above is certainly possible. While attempting to estimate changes in life expectancy from successive censuses is unsafe and backcasting population estimates beyond 10 years produces unreliable and implausible results, it is likely that estimating changes in life expectancy within a 10-year period using backcast population estimates, as carried out by AIHW [33] can provide a useable estimate of trend, even though the levels of life expectancy may be overestimated through the use of data sources each of which is known to under identify Indigenous people.

To its credit, the ABS commissioned an independent review of its Indigenous life expectancy estimate in 2019, which reported in 2021.

Taylor and her colleagues [34] seem to favour the cohort-interpolated approach over the backcasting method for estimating populations and that warrants further investigation. Equally, the Voluntary Indigenous Identifier (VII) on Medicare data may be sufficiently complete to provide an alternative source of identification and it may be appropriate for both ABS and AIHW to consider the potential for using the VII as a tool to reduce under identification in death records.

### 3.2. Identifiers on Private Pathology Request Forms

This issue has been on the national agenda for years but remains unresolved. Similar issues on the inclusion of indigenous identifiers on the records of private hospitals were dealt with decades ago and COVID-19 provides a real opportunity for the issue to be finally rectified along the lines recommended in the 2013 AIHW Report [43].

### 3.3. Community Level Data

“Accurate and locally relevant data on demographics, health outcomes, health determinants and access to services is key to inform decision making by local communities, services and for program and policy evaluations [44].” However, provision of data at small area level presents significant, technical and logistical challenges. The AIHW is developing an Indigenous Community Insights website which will facilitate access to data at a regional level and also produces data for Indigenous Advancement Strategy (IAS) regions and sub regions. The IDN is also focused on provision of regional level data and as Professor Langton said, “By supporting communities and community-controlled organisations to collect their own data and use government-held data, the coalition of peaks and the IDN are helping communities to tell their own stories about what is working for them and what isn’t [22]”.

### 3.4. Measuring Wellbeing

“Accurate wellbeing measures tell us what works and what does not work to improve wellbeing, inform patient and clinical decision making, service delivery, policy, and ultimately improve patient outcomes. The absence of a robust culturally relevant wellbeing measure has significantly hindered progress in improving wellbeing for all Aboriginal and Torres Strait Islander Australians” [45]. This topic is of interest and importance both nationally and internationally [46,47,48,49,50]. Within Australia, Professor Garvey and her colleagues have developed and are testing a nationally relevant instrument to measure the wellbeing of Aboriginal and Torres Strait adults. The measure includes 32 items across 10 dimensions including, for example: Balance and Control, Hope and Resilience, Culture and Country, Spirit and Identity, and Racism and Worries. The research team are developing a short form version of What Matters 2 Adults and have commenced work to develop a What Matters 2 youth wellbeing measure (12–17 years) and are piloting a project to test methods with Aboriginal and Torres Strait Islander children <11 years [46].

### 3.5. Use of Data for Management Purposes

The ground-breaking new National Agreement on Closing the Gap between the Coalition of Aboriginal and Torres Strait Islander Peak Organisations and all Australian governments [23] provides for “Shared access to location specific data and information [that] will support Aboriginal and Torres Strait Islander communities and organisations to support the achievement of the Priority Reforms” through partnership, “making evidence-based decisions on the design, implementation and evaluation of policies and programs for their communities in order to develop local solutions for local issues and “measuring the transformation of government organisations operating in their region to be more responsive and accountable for Closing the Gap”. There is also an acceptance of the desirability of local level data to enable local decision-making, and the need for Aboriginal and Torres Strait Islander communities and organisations to be “supported by governments to build capability and expertise in collecting, using and interpreting data in a meaningful way”.

If translated into action, these agreements would be very important reforms. However, they will not necessarily resolve the fundamental issue, of not just guaranteeing access to data, but using that data at all levels of government, by service providers to improve performance. The failure to fully utilize the data that does exist is a central element in the relative lack of progress in recent years. This is because access to and provision of data is not an end in itself, but an integral element in the policy and planning cycle, where data is used to monitor and improve performance, refine policy, and progressively improve outcomes as set out in the Planning Cycle diagram [51] below (Figure 3).

Information should play a vital role in several of the Actions in the Planning Cycle diagram—in Step 3, Situational Analysis; in Step 4, Review of available resources; and most importantly in Monitoring and Evaluation in Step 11, but at present is not being utilised to anything like its full potential [52,53,54,55,56,57]. Note also that the cycle is just that, a continuous cycle, not a static or periodic process. An essential requirement is to have formal reviews of performance at monthly intervals for service providers, and six monthly and annual reviews involving communities, funders, service providers, and policy makers.

### 3.6. Indigenous Data Governance

There are still many issues that need to be resolved regarding Aboriginal and Torres Strait Islander data in official statistics. Despite the investments in data capabilities in Australia, efforts are still needed to meet the needs and aspirations of Aboriginal and Torres Strait Islander people by facilitating Indigenous Data Sovereignty through Indigenous Data Governance processes. One recommendation is that Aboriginal and Torres Strait Islander people are supported in the development of mechanisms to govern their data. This should be in alignment with current developments in ID-SOV, whereby Indigenous peoples have the right to exercise authority and govern the affairs of the use of Indigenous data that reflects Indigenous peoples interests and aspirations [58]. For Aboriginal and Torres Strait Islander people, this is enacting self-determination in the collection and use of data and acts to redress the existing unequal power distributions currently seen in Australian society. It is important to ensure that Aboriginal and Torres Strait Islander epidemiologists and demographers lead the way in discussions on data collection, quality, and reporting regarding official statistics. This is to enable existing data infrastructures and data systems to work optimally for Aboriginal and Torres Strait Islander people and to ensure there are established mechanisms of expert voice as Aboriginal and Torres Strait Islander communities move closer towards data control and ownership within Australia.

## 4. Conclusions

Much progress has been made in the provision of information but there are a number of immediate challenges—and opportunities. A central lesson of the past is that for information to achieve its potential, it has to be used and used in a way which links policy, funding, implementation, monitoring and evaluation in a continuous policy/planning cycle, and that cycle has yet to be instituted in a systematic way across all levels of service delivery, government and communities. There is now the potential for Aboriginal and Torres Strait Islanders to be not just partners but leaders in the design, collection and use of information, but this also requires a concerted effort to train Aboriginal and Torres Strait Islanders for those tasks and responsibilities.

## Figures and Tables

**Figure 1 ijerph-18-10274-f001:**
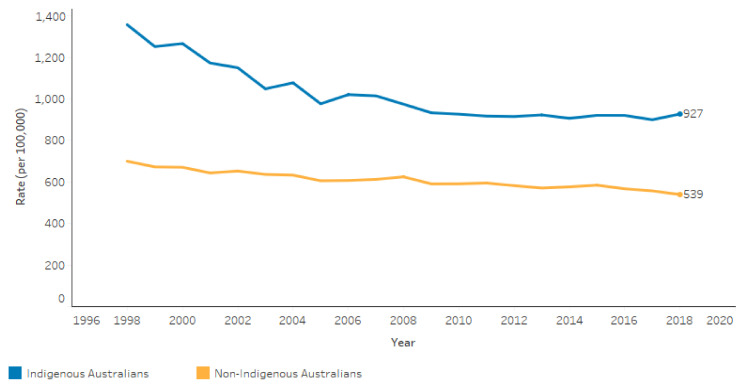
Age-standardized death rates, by Indigenous status, NSW, Qld, WA, SA, and NT, 1998–2018. Source: https://www.indigenoushpf.gov.au/measures/1-22-all-causes-age-standardised-death-rates. (accessed on 16 August 2021).

**Figure 2 ijerph-18-10274-f002:**
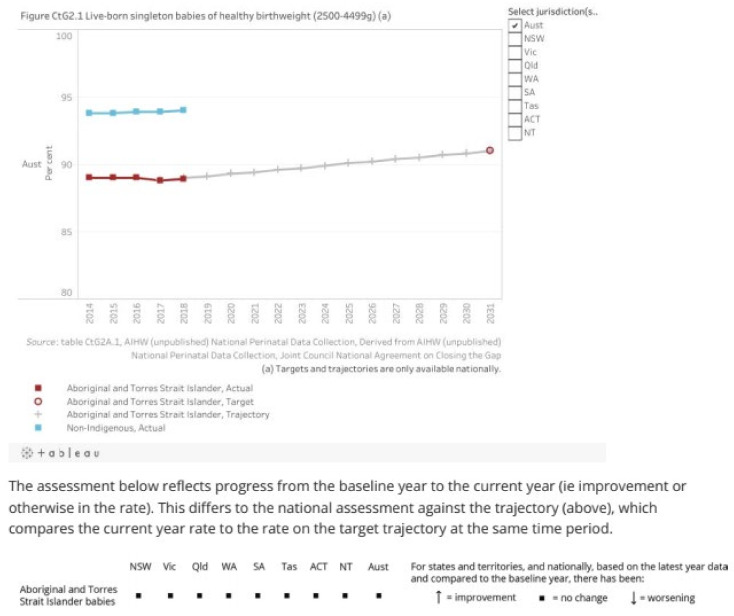
By 2031, increase the proportion of Aboriginal and Torres Strait Islander babies with a healthy birthweight to 91 percent. Nationally in 2018, 88.9 percent of Aboriginal and Torres Strait Islander babies born were of a healthy birthweight. This is similar to 2017 (the baseline year). Source: https://www.pc.gov.au/closing-the-gap-data/annual-data-report/2021/closing-the-gap-annual-data-compilation-report-july2021.pdf (accessed on 16 August 2021).

**Figure 3 ijerph-18-10274-f003:**
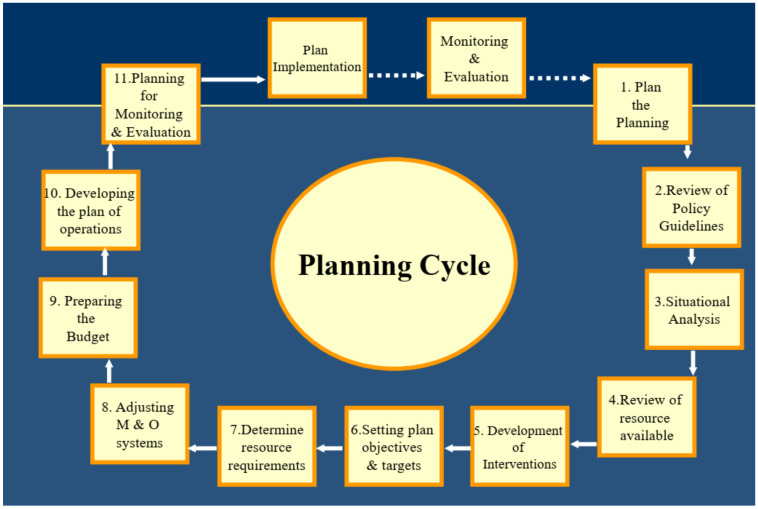
Planning Cycle.

## Data Availability

Not applicable.

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
