# Peer review of "Australian Aboriginal and Torres Strait Islander Health Information: Progress, Pitfalls, and Prospects"

_ijerph, 2021, doi:10.3390/ijerph181910274_

Round 1

Reviewer 1 Report

This is a very informative paper in terms of the historical and contemporary context in Australia for the collection of Aboriginal and Torres Strait Islander health statistics. The authors demonstrate their extensive knowledge of the area, and while remaining largely impartial they do at times reflect their frustrations at inaction - in both the accurate collection and use of health information.

However, I question their central premise that "the failure to fully utilise the data that does exist is a central element in the relative lack of progress in recent years" (551-2). I can see the foundation for this assertion but I do not think the argument supporting it is well enough made. This is largely because the paper is descriptive rather than analytical.

In many places in the paper I found myself wondering 'why?' - why had it taken so long for Aboriginal and Torres Strait Islander peoples to be made visible in health statistics? why had initiatives come and gone? why had governance finally be granted to Aboriginal and Torres Strait Islanders peoples (NAGATSIHID) been taken away again in 2019? This questioning then made the absence of an analytical frame more obvious.

I encourage the authors to be explicit about their approach to the history they describe, and perhaps place it more within the context of calls in Australia and around the world for the voices and concerns of Indigenous peoples to be taken more seriously. The UN Declaration on the Rights of Indigenous People's is a groundbreaking assertion of Indigenous rights and sovereignty that Australia was very slow in signing up to, and yet it clearly signalled a sea-change in how relationships with Indigenous peoples ought to be framed within colonial nation-states like Australia. Such movements - internally and internationally - can help frame up government moves to collect information and plan for Aboriginal and Torres Strait Islander people's health and wellbeing. These moves include the recognition of people as citizens, as fully human, and worthy of investment in their wellness - even if these 'investments' have largely been top-down, assimilatory actions that deny the rights of Aboriginal and Torres Strait Islander peoples to live good, culturally-informed lives.

In addition, to describing the framing or lens for their view of the collection, accuracy and use of health information, I would like the authors to be more explicit about their methodology - how have they gone about analysing the strategies that have been developed and the actions that have been taken? At the moment, the descriptive (rather than analytical) overview of these strategies and actions is deceptively 'objective' and yet the authors clearly have a viewpoint that a methodology might allow them to more easily express. Implementing a methodology would also allow for the synthesis of what is presently just a collection of described events across a timeline, that lacks a connecting narrative.

The groundwork in this paper has been done, as the authors clearly know their material and have a critique of it. What I'm asking for is the next layer, the next analytical step that allows readers to more fully understand what's going in Australia and why? (or at least the authors' analysis of this). I learned a lot from the paper but I came away feeling none-the-wiser, and I wanted to be wiser. I wanted to know what the political, social, economic and historical drivers are and how the changes through time that are described, up to the contemporary expressions of health information, reflect these drivers and other agendas - potentially to the detriment of Aboriginal and Torres Strait Islander people living good lives.

I'd really encourage the authors to take this next step, as this sort of analysis is sorely needed. I also believe that they are more than capable of responding to this request.

Other points

  1. I am wary of the mythology presented in Figure 3, about how planning and implementation happens. If we adhere to this mythology (and deny that things are far more chaotic and random than depicted) then we will always be dissatisfied with the way evidence informs decision-making. 
  2. Accurate wellbeing measures measure wellbeing rather than what works to improve wellbeing. To know what works, evaluative information needs to be added to monitoring data.
  3. I've also done a quick scan of what's going on in Australia to measure Indigenous wellbeing and have found that more is happening that the one initial cited.
  4. While the Indigenous data sovereignty movement is described, there seems some reluctance on the part of the authors to connect sovereignty and governance - or to present an analysis of the difference. It is important to reflect current movements - particularly ones that have taken the world by storm because they resonate with Indigenous peoples - like Indigenous Data Sovereignty.
  5. I found it difficult to track through the long quotes that are included and wonder if all are necessary or if the authors can put more into their own words.

Author Response

We thank the reviewer for the considered and thoughtful comments.

The reasons behind the inadequate and tardy actions by Australian governments on so many aspects of Indigenous life are many and varied. This a complex topic and worthy of a separate paper in its own right. That however is not the focus of this paper which is essentially a review paper describing progress, pitfalls and prospects of Aboriginal and Torres Strait Island health information. 

Nonetheless this is an important point and we have added in 6 references to lines 561 - which provide a historical perspective and throw some light on key issues behind the lack of use of information and tardy and inadequate responses by government. 

The UN Declaration of the Rights of Indigenous Peoples was signed by Australia in 2009 following a change in government. In our view it was symbolic of a change in approach by the new government and its Closing the Gap agenda, without necessarily having a direct and immediate impact itself on issues of health information for Aboriginal and Torres Strait Island people. Many of the steps to improve Aboriginal and Torres Strait island health information long predate the signing of UNDRIP, although in recent times it has had an influence on the data sovereignty movement - see new reference at line 260.

In relation to the specific points:

  1. The diagram presented is a WHO document and typical of standard depictions of the planning cycle. It is not meant to represent what actually happens in practice but to highlight steps which should happen but aren't currently happening. The commentary has been further strengthened to emphasise this point (see lines 558-561).
  2. The statement on wellbeing is a direct quote from a respected Indigenous academic and quotation marks have been added to make that explicit.
  3. Additional references have been included at line 526.
  4.  It is not the intention of the authors to appear reluctant to connect Indigenous Data Sovereignty and Indigenous Data Governance. While these issues are central discussions points in the use of Indigenous data, this review’s purpose is not to present any analysis of ID-SOV/ID-GOV in practice. The authors think that this requires its own, comprehensive discussion in regards to Australian population health data, which cannot be adequately achieved in this review. We have noted the importance of facilitating ID-SOV through ID-GOV for clarity. (L 567-570)
  5. We have given considerable thought to the possibility of shortening or paraphrasing the direct quotes as in the important section on Misleading use of statistics. These sections contest the approaches used by major government agencies and it is our view that the best and fairest approach is to use those agencies' exact words and not to try to paraphrase or abbreviate those words, and that remains our preference. 

Reviewer 2 Report

This is an important and comprehensive paper that will be of great interest to a wide readership. This is well written and presented with an enormous body of knowledge within on manuscript. I have just a couple points for clarification to support readership across disciplines.

  1. Suggest the abstract adds the aim of the paper. I acknowledge this is based on a presentation, a clear aim and scope of the abstract will guide the reader over this comprehensive overview.
  2. The opening presents compelling information that many readers would not be aware of. Can references be added to support these historical account to guide readers.
  3. L132 states "While there are a number of advisory committees within government departments to support decision making regarding Aboriginal and Torres Strait Islander data" Could more information and references be added here unclear to the reader where the current gaps are.
  4. Suggest authors clarify L269 "The Australian Government has invested significantly in its national data capabilities to monitor the progress of the nation’s health through data sharing". This appears in contradiction to L132 on NAGATSIGUD and the abolishment in 2019. Possibly I am misunderstanding it, suggest some details on the investment are included.

Author Response

We thank the reviewer for the helpful comments. In relation to the reviewer's specific points:

  1. The journal indicates that the abstract should be 200 words. The current abstract is a little over that limit and in its present content outlines the key points in the paper . Can we leave this to the editor and if need be, a more explicit description of the aim and scope could be added?
  2. The opening paragraph paraphrases references 1 and 2 which contain the references for the developments described. If need be, those references could be extracted from those two references and added to the rather lengthy reference list but again, can we leave this decision to the editor?
  3. The text has been clarified on this point and additional references included. (L133)
  4. The section on data sharing refers to initiatives for the population as a whole. NAGATSIHID was a national advisory and coordination body to address the many deficiencies in health information for the Aboriginal and Torres Strait Island population. Its abolition did not mean that government agencies could not take individual action, including provision of additional funding, for the population as a whole - or for Indigenous people.